# Mitochondrial Peroxiredoxin 3 Is Rapidly Oxidized and Hyperoxidized by Fatty Acid Hydroperoxides

**DOI:** 10.3390/antiox12020408

**Published:** 2023-02-07

**Authors:** Giuliana Cardozo, Mauricio Mastrogiovanni, Ari Zeida, Nicolás Viera, Rafael Radi, Aníbal M. Reyes, Madia Trujillo

**Affiliations:** 1Departamento de Bioquímica, Facultad de Medicina, Universidad de la República, Montevideo 11800, Uruguay; 2Centro de Investigaciones Biomédicas, Universidad de la República, Montevideo 11800, Uruguay

**Keywords:** peroxiredoxin, mitochondria, fatty acid hydroperoxide, lipid peroxidation, antioxidant systems, kinetics

## Abstract

Human peroxiredoxin 3 (*Hs*Prx3) is a thiol-based peroxidase responsible for the reduction of most hydrogen peroxide and peroxynitrite formed in mitochondria. Mitochondrial disfunction can lead to membrane lipoperoxidation, resulting in the formation of lipid-bound fatty acid hydroperoxides (_L_FA-OOHs) which can be released to become free fatty acid hydroperoxides (_f_FA-OOHs). Herein, we report that *Hs*Prx3 is oxidized and hyperoxidized by _f_FA-OOHs including those derived from arachidonic acid and eicosapentaenoic acid peroxidation at position 15 with remarkably high rate constants of oxidation (>3.5 × 10^7^ M^−1^s^−1^) and hyperoxidation (~2 × 10^7^ M^−1^s^−1^). The endoperoxide-hydroperoxide PGG_2_, an intermediate in prostanoid synthesis, oxidized *Hs*Prx3 with a similar rate constant, but was less effective in causing hyperoxidation. Biophysical methodologies suggest that *Hs*Prx3 can bind hydrophobic structures. Indeed, molecular dynamic simulations allowed the identification of a hydrophobic patch near the enzyme active site that can allocate the hydroperoxide group of _f_FA-OOHs in close proximity to the thiolate in the peroxidatic cysteine. Simulations performed using available and herein reported kinetic data indicate that *Hs*Prx3 should be considered a main target for mitochondrial _f_FA-OOHs. Finally, kinetic simulation analysis support that mitochondrial _f_FA-OOHs formation fluxes in the range of nM/s are expected to contribute to *Hs*Prx3 hyperoxidation, a modification that has been detected in vivo under physiological and pathological conditions.

## 1. Introduction

In most mammalian cells, mitochondria are main sources of oxidizing species [1,2]. Among them, it forms different hydroperoxides such as hydrogen peroxide (H_2_O_2_), peroxynitrite and either free or lipid-bound fatty acid hydroperoxides (_f_FA-OOHs or _L_FA-OOHs, respectively), many of which participate in cellular redox signaling pathways at low, physiological concentrations [3,4,5]. However, increased steady-state levels of these species can lead to mitochondrial dysfunction, cytotoxicity and several disease states [4,6,7,8,9,10]. Different peroxidases reduce hydroperoxides in mitochondria [11]. Peroxiredoxin 3 (Prx3) catalyzes the reduction of most of mitochondrial H_2_O_2_ [12]. Furthermore, our previous work indicates that Prx3 is an efficient peroxynitrite reductase that represents, together with peroxiredoxin 5 (Prx5), the main target for peroxynitrite in mitochondria [13]. Both H_2_O_2_ and peroxynitrite can lead to Prx3 inactivation due to peroxidatic cysteine (Cys_P_) hyperoxidation to sulfinic acid (RSO_2_H) or even to sulfonic acid (RSO_3_H), although the enzyme is considered to be less susceptible to this modification than other mammalian typical two cysteine peroxiredoxins (Prxs) such as Prx2 [13,14]. This was ascribed to a slower rate constant of resolution, i.e., change from the fully folded (FF) to locally unfolded (LU) conformation followed by the formation of an inter-subunit disulfide with the so-called resolving cysteine residue (Cys_R_), in the case of Prx3 [13,15].

_L_FA-OOHs can be produced non-enzymatically due to mitochondrial membrane lipoperoxidation processes, initiated by the reaction of different one-electron oxidants with polyunsaturated fatty acids (PUFA) such as linoleic acid (LA) and arachidonic acid (AA). Alternatively, enzymatic mechanisms can also lead to FA-OOHs formation [16,17]. In particular, the inner mitochondrial membrane is characterized by a high degree of unsaturation and is rich in phosphatidyl ethanolamine, phosphatidyl choline and cardiolipin, a phospholipid in which two phosphatidic acid groups are attached to a central glycerol molecule (Membrane composition varies among cell compartments and tissues. In rat liver inner mitochondrial membranes, phosphatidyl ethanolamine, phosphatidyl choline and cardiolipin represent 40%, 40% and 20% of total phospholipids, respectively [18]). The interaction of cardiolipin with cytochrome *c* leads to an increase in the peroxidase activity of the latter that promotes the oxidation of PUFA in the phospholipid [19]. Increased formation of oxidants and lipid peroxidation induces the activation of mitochondrial phospholipases, resulting in the release of the corresponding hydroperoxides (_f_FA-OOHs) from oxidized membranes, which participate in physiological signaling cascades and in mitochondrial uncoupling [16,20,21,22]. Additionally, _f_FA-OOHs formed in the cytosol can diffuse into the mitochondria leading to changes in the membrane potential, release of cytochrome *c* and generation of further oxidative species [23]. Increased lipoperoxidation is a key participant in ferroptosis, an iron-dependent form of regulated cell death characterized by increased lipoperoxidation in cellular membranes [24] that may result in pathologies such as neurodegenerative diseases [25,26].

Both free and lipid-bound FA-OOHs are reduced to the corresponding alcohols by the selenocysteine-containing enzyme glutathione peroxidase 4 (GPx4) [27,28]. Indeed, homozygous GPx4 knock-out mice are not viable and cells lacking GPx4 activity die due to increased lipoperoxidation that is a hallmark of ferroptosis [29]. Although the enzyme is expressed as cytosolic, mitochondrial and nuclear isoforms, inactivation of mitochondrial GPx4 alone allowed normal embryogenesis and postnatal development but caused male infertility. This is consistent with the low expression of the mitochondrial isoform in most somatic tissues and its high abundance in spermatozoa [30,31]. Furthermore, mitochondrial GPx4 is essential for the development and survival of photoreceptors in vivo [32]. Mitochondria also expresses GPx1, present in various cell compartments and widely distributed in cellular tissues, which rapidly reduces H_2_O_2_, peroxynitrite and _f_FA-OOHs, but is unable to reduce complex _L_FA-OOHs [33]. The ability of some members of the Prx family to interact with phospholipid membranes and to reduce FA-OOHs was firstly demonstrated by Cha et al. in 2000 [34] who reported that human Prx2 rapidly reduces _f_FA-OOH and interacts with erythrocyte plasma membrane through a C-terminal region [34]. Mammalian Prx6 also reduces FA-OOHs forming part of phospholipids in membranes [35]. Some other Prxs reduce _f_FA-OOHs but have no- or only a very weak activity with _L_FA-OOHs [36,37,38]. In some cases, _f_FA-OOHs not only oxidize but also hyperoxidize Cys_P_ [39,40]. The molecular basis for the oxidizing substrate specificity in Prxs is only starting to be unraveled [41]. Some previous data suggest that Prx3 might be able to catalyze the reduction of _f_FA-OOHs. For example, cells lacking Prx3 showed an increased susceptibility of the tumor suppressor protein PTEN, a member of the protein tyrosine phosphatase family, to be oxidized by a _f_FA-OOH [42]. Furthermore, Prx3 got hyperoxidized in cells exposed to AA-derived hydroperoxides [43]. However, since addition of _f_FA-OOHs to cells can lead to mitochondrial dysfunction and further oxidant production [44], a direct reaction of FA-OOHs and Prx3 remains to be investigated. Although the enzyme is mostly expressed in mitochondria, it can also be located in cell surfaces [45]. For instance, it was detected in membrane-enriched fractions from cancer cell lines exposed to RSL3, an inhibitor of GPx4 that causes ferroptosis. However, Prx3 siRNA failed to demonstrate a protective role against this form of death in those cells [46]. Interestingly, mouse embryonic fibroblast expressing a mutated form of GPx4 (Sec to Cys) showed a compensatory up-regulation of Prx3 [47]. Furthermore, overexpression of Prx3 ameliorates the symptoms of mild but not severe forms of ferroptosis of spinal motor neurons from GPx4 neuron inducible knockout mice [48].

Herein, we investigated the ability of human Prx3 (*Hs*Prx3) to catalyze the reduction of FA-OOHs. In particular, we investigated the reactions of *Hs*Prx3 with _f_FA-OOHs involved in eicosanoid synthesis, i.e., derived from 15-lipoxygenase-catalyzed peroxidation of AA and EPA (15S-hydroperoxy-5Z,8Z,11Z,13E-eicosatetraenoic acid (15(S)-HpETE) and 15S-hydroperoxy-5Z,8Z,11Z,13E,17Z-eicosapentaenoic acid (15(S)-HpEPE), respectively) and derived from cyclooxygenase-catalyzed peroxidation of AA (the endoperoxide-hydroperoxide prostaglandin G_2_ (PGG_2_)), which is a precursor for the 2-series prostaglandin and thromboxane synthesis (structures are shown in Appendix A). Our data indicate that the enzyme is very rapidly oxidized as well as hyperoxidized by _f_FA-OOHs, alerts on the interpretation of kinetic results obtained by different methodologies and could help to explain at least in part Prx3 hyperoxidation detected in mammalian tissues [49,50].

## 2. Materials and Methods

### 2.1. Reagents

Dithiotreitol (DTT), N-ethylmaleimide (NEM), beta-mercaptoethanol (β-ME), reduced nicotinamide adenine dinucleotide phosphate (NADPH), isopropyl-β-d-thiogalactopyranoside (IPTG), kanamycin sulphate, ampicillin, 2-iodoacetamide, diethylenetriaminepentaacetic acid (DTPA), 8-anilino-1-naphtalene sulfonic acid (ANS) and imidazol were from Sigma-Aldrich (Darmstadt, Germany). H_2_O_2_ was obtained from Mallinckrodt Chemicals (St. Louis, MO, USA). 15(S)-HpETE (≥98% pure), 15(S)-HpEPE (≥98% pure), 9α,11α-epidioxy-15S-hydroperoxy-prosta-5Z,13E-dien-1-oic acid (PGG_2_, ≥95% pure), arachidonic acid (AA, ≥98% pure) and eicosapentaenoic acid (EPA, ≥98% pure) (Appendix A) were obtained from Cayman Chemical (Ann Arbor, MI, USA) as ethanolic solutions (or in acetone solution in the case of PGG_2_) and stored under argon at −80 °C. All other reagents were obtained from standard commercial sources and used as received. Experiments were performed in 50 mM phosphate buffer plus 0.1 mM DTPA, pH 7.8 and 25 °C, except otherwise indicated.

### 2.2. Hydroperoxide Concentration Measurements 

The concentration of H_2_O_2_ was determined from absorbance measurements at 240 nm (ε = 43.6 M^−1^cm^−1^) [51].

Hydroperoxide consumption by *Hs*Prx3 under non-catalytic conditions was measured mixing reduced enzyme with the hydroperoxide of interest and determining remaining hydroperoxide concentrations after 30 s of incubation by the ferrous oxidation–xylenol orange method (FOX assay) [39,52]. The extinction coefficients for the different hydroperoxides using this assay were 15(S)-HpEPE, Abs_560_ = 60,000 M^−1^cm^−1^; PGG_2_, Abs_560_ = 44,000 M^−1^cm^−1^; H_2_O_2_, Abs_560_ = 43,000 M^−1^cm^−1^, which are in close agreement with previously reported values [39,53]. 

### 2.3. Enzyme Expression and Purification

Recombinant *Hs*Prx3 (wild type, wt*Hs*Prx3 as well as a Cys_P_ to Ser mutated form, C_P_S*Hs*Prx3) lacking the mitochondrial targeting sequence and containing three additional N-terminal Gly residues were expressed in BL21 Star (DE3) *Escherichia coli* strain and purified as previously reported [13]. Human thioredoxin 1 (*Hs*Trx1), which acts as an efficient reducing substrate for *Hs*Prx3 in vitro [54], was expressed as in [55]. The plasmid for recombinant expression of *Echinococcus granulosus* thioredoxin glutathione reductase lacking the glutaredoxin (Grx) domain (*Eg*TR) was obtained from Mariana Bonilla and Lucía Turell, from Institut Pasteur Montevideo and Universidad de la República, Montevideo, Uruguay, respectively, and was expressed and purified as previously [56].

Protein concentrations were measured at 280 nm using extinction coefficients of 19,940 M^−1^cm^−1^ per reduced *Hs*Prx3 subunit and 7100 M^−1^cm^−1^ for *Hs*Trx1 as previously [13]. *Eg*TR protein concentration was measured as in [56]. 

### 2.4. Reduction of HsPrx3

In some experiments, *Hs*Prx3 was reduced immediately before use by incubating the enzyme with DTT (2 mM) for 30 min on ice. Excess reductant was removed by gel filtration using a HiTrap desalting column (Amersham Biosciences, Amersham, UK) and freshly made degassed sodium phosphate buffer 50 mM plus 0.1 mM DTPA, pH 7.8. 

### 2.5. Catalytic Activity of HsPrx3

The catalytic activities of *Hs*Prx3 using H_2_O_2_ or different _f_FA-OOHs as oxidizing substrates were measured using a coupled assay following NADPH oxidation at 340 nm (ε_340 nm_ = 6220 M^−1^s^−1^) (150 µM) [57] in the presence of *Eg*TR (40 nM), *Hs*Trx1 (30 µM) and *Hs*Prx3 (0.8 µM) in sodium phosphate buffer 50 mM plus 0.1 mM DTPA at pH 7.8 and 25 °C. The mixture was pre-incubated during a minute and the reaction was started by adding 20 µM of the indicated hydroperoxide, rapidly mixed in a vortex and transferred to a cuvette to start measurements. In order to determine whether the enzyme remained active after the incubation with the hydroperoxide of interest, H_2_O_2_ (20 µM) was added 3 min after the initial incubation. Control experiments using different concentrations of *Eg*TR and *Hs*Prx3 were performed to ensure that reduction of *Hs*Trx1 by *Eg*TR was not rate limiting the oxidation of NADPH by the hydroperoxides of interest in the presence of *Eg*TR, *Hs*Trx1 and *Hs*Prx3. *Eg*TR, which as mammalian TR is a selenium-containing protein, was used due to its availability as a highly pure recombinant protein in our lab. Measurements were performed using a Shimadzu UV-Vis spectrophotometer (Shimadzu, Kyoto, Japan) equipped with a Peltier-based temperature control. 

### 2.6. Kinetics of HsPrx3 Oxidation and Hyperoxidation by _f_FA-OOHs

As in other members of the typical two-cysteine Prx family [58,59,60], changes in the redox state of Cys_P_ of *Hs*Prx3 are accompanied by changes in the tryptophan-dependent intrinsic fluorescence intensity of the protein [13]. The oxidation of the thiol group in *Hs*Prx3 Cys_P_ to sulfenic acid causes a rapid decrease in intrinsic fluorescence while resolution causes a slower phase of fluorescence increase. Since oxidation is a bimolecular process, the observed rate constant of the decrease in intrinsic fluorescence depends on the concentration of the oxidant. On the contrary, resolution is an intramolecular process and its rate constant is independent on the oxidant concentration. Hyperoxidation can compete with resolution leading to an acceleration of the second phase of fluorescence change (Appendix A). We therefore took advantage of this fluorescence behavior of the protein to measure the kinetics of *Hs*Prx3 Cys_P_ oxidation and hyperoxidation. Reduced wt or C_P_S*Hs*Prx3 in 50 mM sodium phosphate buffer plus 0.1 mM DTPA pH 7.8 in one syringe was rapidly mixed with the indicated concentrations of the hydroperoxide of interest in the same buffer from a second syringe at 11–12 °C, and the protein intrinsic fluorescence changes were recorded using a SX20 Applied Photophysics stopped-flow spectrofluorimeter (Applied Photophysics, Leatherhead, UK) (λ_ex_ = 295 nm, total emission). Data were fitted by exponential functions to obtain the observed rate constants of each phase (*k*_obs_). The rate constants of the reaction of *Hs*Prx3 Cys_P_ oxidation and hyperoxidation were calculated from the slope of the plot of the *k*_obs_ of the decrease phase and the increase phase of fluorescence change, respectively, plotted versus hydroperoxide concentration. 

### 2.7. SDS PAGE and Western Blot Analysis of HsPrx3 Hyperoxidation

Reduced *Hs*Prx3 (5 µM) was treated with the indicated concentrations of H_2_O_2_ or _f_FA-OOHs in sodium phosphate buffer 50 mM pH 7.8 plus 0.1 mM DTPA. After 15 min, the alkylating agent NEM (20 mM) was added. Samples were resolved in 15% SDS-PAGE electrophoresis under non-reducing (−β-ME) or reducing (+β-ME) conditions and were stained using Coomasie. Alternatively, samples were transferred to a nitrocellulose membrane to detect *Hs*Prx3 Cys_P_ hyperoxidation to sulfinic/sulfonic (Cys_P_-SO_2/3_H) acid using a commercial rabbit polyclonal antiPrx-Cys_P_-SO_2/3_H antibody (Abcam, 1b16830, Waltham, MA, USA). Secondary anti-rabbit antibodies were IR-Dye 800CW and data were acquired using an Odyssey imaging system from LI-COR Bioscience (Lincoln, NE, USA).

### 2.8. Mass Spectrometry Analysis of HsPrx3 Modifications by FA-OOHs

Reduced *Hs*Prx3 (10 µM) in 50 mM phosphate buffer plus 0.1 mM DTPA (pH 7.8, 25 °C) without further treatment, exposed to 15(S)-HpEPE (10 or 20 µM) or to H_2_O_2_ (20 µM) were alkylated with 2-iodoacetamide (5 mM) for 30 min at 37 °C. The excess alkylating agent was removed by gel filtration using PD SpinTrap G-25 columns (GE Healthcare, Chicago, IL, USA) equilibrated with 20 mM ammonium bicarbonate buffer (pH 7.4, 25 °C). All samples were loaded into a C4 column (214MS5115, Grace Vydac, Hesperia, CA, USA) for HPLC separation. Mobile phase consisted of 0.1% formic acid in nanopure water (solvent A) and 0.1% formic acid in CH_3_CN (solvent B), and elution of the protein was performed with a 10-min linear gradient of solvent B (5–50%) followed by an additional 10 min at 50% solvent B at 100 μL/min. An ESI-triple quadrupole mass spectrometer (QTRAP4500, ABSciex Framingham, MA, USA) was employed for detection. The spectrometer was set in Q1 positive mode in the 500–2000 *m*/*z* range with a scan rate of 200 Da/s and Q1 resolution in unit. Parameters used were as follows: Ion Sray voltage, 5000 V; Ion Source Temperature, 300 °C; Declustering Potential, 120 V; Entrance Potential, 10; Curtain Gas, 20 psi; Gas1, 30 psi; Gas2, 20 psi. Data acquisition was performed using Analyst 1.6.2 (ABSciex) and PeakView 2.2 (ABSciex) software was used for data analysis and deconvolution of all spectra.

### 2.9. Fatty Acids Binding to HsPrx3

The ability of *Hs*Prx3 to bind fatty acids (FAs) was evaluated by two different experimental methodologies. Firstly, we analyzed the interactions of *Hs*Prx3 with ANS, a fluorescent compound used as a probe of protein surface hydrophobicity in the proximity of cationic groups [61,62]. ANS binding to this protein microenvironment leads to a blue shift and an increase in the probe quantum yield. Furthermore, due to its spectral properties ANS could lead to intrinsic, tryptophan-dependent protein fluorescence resonance energy transfer (FRET) [63]. Fluorescence measurements were performed using a Jasco FP-8500 spectrofluorimeter and 500 µL, 10 mm path length quartz cuvettes. For direct fluorescence measurements, excitation was performed at 375 nm and emission spectra (440–600 nm) were obtained. *Hs*Prx3 (5 µM) was titrated with ANS in sodium phosphate buffer 50 mM pH 7.8 plus 0.1 mM DTPA. Control experiments in the absence of *Hs*Prx3 were also performed. In FRET experiments, *Hs*Prx3 (2 µM, 5 µM and 8 µM) was excited at 295 nm, and emission spectra were recorded between 315 nm and 550 nm. Binding constants were obtained fitting the plot of change in intrinsic fluorescence intensity versus ANS concentration to hyperbolic functions as well as through Scatchard analysis.

Additionally, heat unfolding experiments of *Hs*Prx3 in the absence and presence of AA were performed to analyze the effect of the FA on the thermal stability of the protein. Briefly, the tryptophan-dependent intrinsic fluorescence intensity of *Hs*Prx3 (4 µM) (λ_ex_ = 295 nm, λ_em_ = 335 nm) in the absence or presence of AA (10 or 20 µM) plus an excess of DTT (2 mM) was registered in a Jasco FP-8500 spectrofluorimeter (Jasco, Tokyo, Japan), while an increase in temperature (35–70 °C) with a heating rate of 1.5 °C/min was applied to the sample. Sample buffer was 50 mM phosphate buffer plus 0.1 mM DTPA, pH 7.8 and a 10 mm path length quartz cuvette (500 µL) was used. The apparent melting temperatures (Tm) as midpoint of the denaturation curves were obtained fitting the data to a sigmoidal function.

### 2.10. Computer-Assisted Molecular Modeling of FA Binding to HsPrx3

To perform docking simulations of the binding of 15(S)-HpETE to reduced *Hs*Prx3, a dimeric unit made of chains A and B was extracted from its reported crystal structure PDB id 5jcg and subjected to a short molecular dynamic [64] simulation [64] as previously reported by our group [40,41]. Briefly, the dimer was solvated using a default method, with an octahedral box of 12 Å in radius with TIP3P water molecules [65]. Protein residue parameters correspond to the parm14SB Amber force field [66]. MD was performed using periodic boundary conditions with a 10 Å cutoff and particle mesh Ewald (PME) summation method for treating the electrostatic interactions. The hydrogen bond lengths were kept at their equilibrium distance by using the SHAKE algorithm, while temperature and pressure were kept constant with a Langevin thermostat and barostat, respectively, as implemented in the AMBER program (Amber, University of California, San Diego, CA, USA) [67]. The system was optimized in 1000 steps (10 with steep gradient and the rest with conjugate gradient). Then, it was slowly heated from 0 K to 300 K for 20 ps at constant pressure, with Berendsen thermostat, and pressure was equilibrated at 1 bar for 5 ps. After these two steps, a 10 ns long MD simulation at constant temperature (300 K) and constant volume was performed, followed by a 200 ns long trajectory. The last structure resulting from the MD was used for the molecular docking of 15(S)-HpETE. The structure of the enzyme was considered rigid while 15(S)-HpETE was allowed to be flexible. 500 binding modes were generated and analyzed using standard protocols of AutoDock4.2.6 (The Scripps Research Institute, La Jolla, CA, USA) [68]. 

### 2.11. Simulations of HsPrx3 Hyperoxidation by Fluxes of Mitochondrial Hydroperoxides in the Presence of Peroxidases

Simulations were performed using COPASI 4.37 program. Rate constants, reactant concentrations and formation rates of H_2_O_2_ and _f_FAOOH are indicated in the text. 

## 3. Results

### 3.1. HsPrx3 Is Rapidly Inactivated by _f_FA-OOHs

*Hs*Prx3 catalyzed the reduction of H_2_O_2_ (20 µM) as detected by a decrease in NADPH concentration in the coupled assay described in Methods. No consumption of NADPH was observed in the absence of *Hs*Prx3 (not shown). The reaction ended due to oxidizing substrate depletion and not because of enzyme inactivation due to hyperoxidation, since a new addition of H_2_O_2_ to the cuvette caused a new phase of NADPH consumption of similar rate. This is consistent with previous data indicating that the rate constant of *Hs*Prx3 hyperoxidation by H_2_O_2_ is 1.1 × 10^3^ M^−1^s^−1^ versus a rate constant of resolution of 2 s^−1^ at pH 7.8 and 25 °C [13], which indicates that at 20 µM H_2_O_2_ only ~1% of the enzyme would be hyperoxidized in the first catalytic cycle, a percentage/cycle that should decrease as the oxidant gets consumed. On the contrary, very slow rates of NADPH consumptions were observed when 15(S)-HpETE or 15(S)-HpEPE were used as substrates in the coupled assay. The addition of H_2_O_2_ (20 µM) to the cuvette at the end of the reaction failed to cause a decrease in NADPH concentration, indicating that *Hs*Prx3 was inactive (Figure 1). The rate of NADPH consumption was modest in the case of PGG_2_ (Table 1).

### 3.2. Kinetics of HsPrx3 Oxidation by _f_FA-OOH

The addition of 15(S)-HpETE (Figure 2A), 15(S)-HpEPE (Figure 2B), and PGG_2_ (Figure 2C) in excess to reduced *Hs*Prx3 caused a very rapid decrease in the enzyme intrinsic fluorescence (phase of oxidation of the thiolate in Cys_P_ to sulfenic acid), followed by a slower phase of fluorescence increase (resolution plus hyperoxidation phase) (Appendix A). In the case of 15(S)-HpETE and 15(S)-HpEPE, the first phase was so rapid that it was complete in a few ms even at the lower concentrations of oxidant tested (Figure 2A,B). A half-life of oxidation reaction of <10 ms for _f_FA-OOHs concentrations of 1.5–2 µM as obtained for both 15(S)-HpETE and 15(S)-HpEPE is consistent with rate constants of >3.5 × 10^7^ M^−1^s^−1^ at pH 7.8 and 12 °C. In the case of PGG_2_, the oxidation phases were slower and fitted to exponentials. From the plot of *k*_obs_ of the first phase versus the concentration of PGG_2_, the rate constant of *Hs*Prx3 oxidation by this _f_FA-OOH was obtained as (2.4 ± 0.4) × 10^7^ M^−1^s^−1^ (Figure 2C inset). From the slopes of the plots of *k_obs_* of the second phase of fluorescence change versus the concentration of 15(S)-HpETE, 15(S)-HpEPE and PGG_2_ the rate constants of hyperoxidation were obtained as (1.7 ± 0.1) × 10^7^, (2.6 ± 0.4) × 10^7^ and (3.1 ± 0.7) × 10^5^ M^−1^s^−1^, respectively, (Figure 2D). As controls, the addition of the non-peroxidized fatty acid EPA (3 µM) to reduced *Hs*Prx3 (0.3 µM) caused no change in the protein intrinsic fluorescence. Similarly, the addition of PGG_2_ to reduced C_P_S*Hs*Prx3 (0.3 µM) caused no change in protein intrinsic fluorescence, consistent with the requirement of both Cys_P_ in *Hs*Prx3 and the hydroperoxide group in the fatty acid for the reaction to occur (Figure 2E,F).

### 3.3. Consumption of Hydroperoxides by HsPrx3 under Non-Catalytic Conditions

When H_2_O_2_ (50 µM) was mixed with reduced *Hs*Prx3 (20 µM) a fast consumption of 0.8 equivalents of hydroperoxide by *Hs*Prx3 occurred. Similarly, a 1:1 consumption of PGG_2_/*Hs*Prx3 was obtained. However, the stoichiometry of hydroperoxide consumption was higher (1.7:1) for 15(S)-HpEPE. This is consistent with a fast hyperoxidation process in the case of 15(S)-HpEPE that consumes a second equivalent of oxidant (Figure 3). 

### 3.4. HsPrx3 Is Hyperoxidized at Low Concentrations of _f_FA-OOHs

#### 3.4.1. SDS PAGE and Western Blot Analysis

Reduced *H*sPrx3 (5 µM) without any further addition in the absence of β-ME (Appendix A) was mostly monomeric although it contained a minor proportion of oxidized to disulfide dimeric enzyme that reverted to the monomeric form in the presence of β-ME (Appendix A), consistent with the 0.8 H_2_O_2_/*Hs*Prx3 stoichiometry of hydroperoxide consumption shown in Figure 3. The addition of H_2_O_2_ (500 µM, lanes 3 and 4) or 15(S)-HpETE (10 µM, lanes 5 and 6) to reduced *Hs*Prx3 (5 µM) caused an increase in the dimeric form of the enzyme in the samples without β-ME (lanes 3 and 5) that were fully reverted to the monomeric form in the presence of the reductant (lanes 4 and 6). The presence of monomeric form of the enzymes in samples treated with excess hydroperoxides and without β-ME are consistent with the formation of hyperoxidized monomers as confirmed by Western Blot (see below). 

When reduced *Hs*Prx3 (5 µM) was exposed to H_2_O_2_, hyperoxidation of Cys_P_ was only marginally detected unless excess oxidant was used. As reported [14], hyperoxidation by H_2_O_2_ occurred firstly in one of the subunits of the dimer, while the other was forming an inter-subunit disulfide which is detected as a hyperoxidized dimer, and only at concentrations of H_2_O_2_ ≥ 500 µM hyperoxidized monomers, i.e., subunits with both Cys_P_ hyperoxidized, were detected (Figure 4A). On the contrary, both 15(S)-HpETE and 15(S)-HpEPE caused *Hs*Prx3 Cys_P_ hyperoxidation at lower concentrations, even equimolar, that was detected in both the monomeric and dimeric forms of the protein, in agreement with a fast kinetics of hyperoxidation. PGG_2_-mediated hyperoxidation profile was similar to that of H_2_O_2_, leading to hyperoxidized dimers at concentrations ≤50 µM (not shown).

#### 3.4.2. Mass Spectrometry Analysis

In order to confirm the hyperoxidation of *Hs*Prx3 treated with _f_FA-OOHs, MS analysis was performed. The predicted molecular weight of the reduced *Hs*Prx3 subunit is 21,797 Da (13). Treatment with excess 2-iodoacetamide caused alkylation of reduced *Hs*Prx3 (major peaks of + 113 and +170 which corresponds to the covalent addition of 2 and 3 carbamidomethyl (CAM) groups (+57 each), respectively, Figure 5A and Table 2). The presence of a small fraction of a form of the protein modified by 4 CAM groups in the reduced enzyme that has three Cys residues in its sequence is consistent with the reported side reaction of the alkylating agent with N-terminal as well as side chain amine groups [69]. When reduced *Hs*Prx3 was exposed to H_2_O_2_ (1:2 concentration ratio) and then alkylated, the main products detected were the dimeric oxidized disulfide form of the enzyme as well as alkylated derivatives (Figure 5B and Table 2). However, when reduced *Hs*Prx3 was treated with a 1:1 concentration ratio of 15(S)-HpEPE and then treated with iodoacetamide, the main products had an increase in mass of +89, consistent with CAM addition (+57) plus hyperoxidation (+32) (Figure 5C and Table 2). When the concentration of 15(S)-HpEPE was further increased (2:1 concentration ratio) the major species detected had a mass increment of +104, consistent with CAM addition plus the addition of 3 oxygen atoms (Figure 5C and Table 2). This +104 modification was observed in the monomeric and also in the dimeric form of the enzyme exposed to 15(S)-HpEPE (Table 2). The latter is ascribed to Cys_P_ hyperoxidation to sulfonic acid (+48), since similar addition of 2:1 concentration of 15(S)-HpEPE to C_P_S*Hs*Prx3 yielded a +57 form, consistent with alkylation of the enzyme as the major product (Figure 5D and Table 2). 

### 3.5. Biophysical Experiments Suggest the Binding of Hydrophobic Compounds to HsPrx3

#### 3.5.1. Binding of ANS and AA to HsPrx3

The addition of increasing concentrations of ANS to *Hs*Prx3 (5 µM) in sodium phosphate buffer 50 mM, pH 7.8 plus 0.1 mM DTPA, showed an increase in ANS quantum yield (λ_ex_ = 375 nm) and a blue shift when compared with equal concentrations of ANS in the same buffer in the absence of the enzyme (Figure 6A). Additionally, ANS caused a decrease in tryptophan-dependent *Hs*Prx3 fluorescence emission at 339 nm (λ_ex_ = 295 nm) (Figure 6A) consistent with FRET, as previously reported for several proteins that bind ANS, including the peroxiredoxin AhpE from *Mycobacterium tuberculosi*s [41]. Relative intrinsic fluorescence change (1 − F/F_0_) exhibited a hyperbolic behavior when plotted against ANS concentration, resulting in saturation curves from which the value of the dissociation constant (K_d_) of 65 ± 4 µM and number of binding sites (n) of 1.34 ± 0.05 were obtained. The inset in Figure 6B shows a Scatchard linealization of the data obtained at 2 µM *Hs*Prx3, which yielded similar parameters. The addition of the saturated fatty acid palmitic acid at concentrations below its CMC (60 µM) [70] did not compete with the binding of ANS to the enzyme (not shown).

The apparent melting temperature (Tm) of reduced *Hs*Prx3 (4 µM) increased in the presence of AA (10 µM) and did not further increase in the presence of higher AA concentrations (20 µM), with Tm values of 64.6 ± 1 °C for *Hs*Prx3, 68.1 ± 1 °C for *Hs*Prx3 + AA (10 µM) and 67.2 ± 0.3 °C for *Hs*Prx3 + AA (20 µM) (AA critical micelle concentration ranges between 10–60 μM [71,72]). Addition of similar concentrations of FA solvent (ethanol) caused no change in protein Tm value (Appendix A). This ~4 °C shift is consistent with the ability of *Hs*Prx3 to interact with AA [73].

#### 3.5.2. Modeling 15(S)-HpETE Binding to HsPrx3

The *Hs*Prx3 dimeric unit arising from MD simulations using PDBid 5jcg chains A and B as starting point presents a hydrophobic patch close to Cys_P_ with a solvent accessible surface area value of 256 ± 26 Å^2^ (Figure 7A). This region presents hydrophobic residues from both chains of the dimer, namely: Y39, P40, F43, F45, L142, P143, V167′ and P181’. The lowest energy binding modes of 15(S)-HpETE obtained by molecular docking simulations use this hydrophobic region for binding as shown in Figure 7B. Among these conformations, the most populated best ranked cluster presents the hydroperoxide group of 15(S)-HpETE in proximity to the thiolate of Cys_P_ (Figure 7C). Those were also the structures of overall lower energy, indicating that the hydroperoxide group of the _f_FA-OOH may also contribute to substrate pose.

## 4. Discussion

*Hs*Prx3 was almost inactive in catalyzing the reduction of 20 µM 15(S)-HpETE and 15(S)-HpEPE by *Hs*Trx1 after 15 s required for appropriate mixing for measurements in a conventional spectrophotometer. Moreover, the enzyme lost its activity towards H_2_O_2_ after being treated with those _f_FA-OOHs (Figure 1 and Table 1). However, the enzyme consumed 1.7 15(S)-HpEPE per reduced *Hs*Prx3 under non-catalytic conditions (Figure 3), indicating that the lack of enzymatic activity shown in Figure 1 could be due to a rapid inactivation caused by hyperoxidation of Cys_P_. In the case of H_2_O_2_, in which hyperoxidation should not compete with resolution at low oxidant concentrations (see below) the stoichiometry was 0.8 hydroperoxide consumed per *Hs*Prx3. Furthermore, one PGG_2_ was consumed per *Hs*Prx3 (Figure 3), in agreement with its lower inactivation by this oxidant under catalytic conditions (Table 1) compared with 15(S)-HpEPE. The fact that reduced *Hs*Prx3 consumed less than one H_2_O_2_ per protein reflects a minor fraction of oxidized enzyme (~20%) in our sample after reduction and excess DTT removal (Appendix A) due to the presence of adventitial H_2_O_2_ in buffer solutions as already reported [74].

The addition of equimolar or higher concentrations of 15(S)-HpETE or 15(S)-HpEPE to *Hs*Prx3 caused hyperoxidation of Cys_P_ as detected by Western Blot analysis (Figure 4). In consonance, MS confirmed hyperoxidation to Cys_P_ sulfinic acid even when added at equimolar concentration with respect to the enzyme. Additionally, in the presence of excess 15(S)-HpEPE, *Hs*Prx3 Cys_P_ was not only oxidized to sulfinic acid but also formed a +48 derivative that was not detected in the C_P_S mutated form of the enzyme, which is therefore consistent with sulfonic acid formation, and that appears in the monomeric as well as in the dimeric species (Figure 5). The formation of sulfonic acid in Cys_P_ of *Hs*Prx3 had already been reported in *Hs*Prx3 not only in vitro but also in vivo under physiological conditions*,* in the mitochondria from old rats [49]. 

The rate constants of oxidation of *Hs*Prx3 were similar for every _f_FA-OOHs tested so far (Figure 2), and to those reported for H_2_O_2_- and peroxynitrite-mediated oxidations (Table 3). However, the rate constants of *Hs*Prx3 hyperoxidation differed in several orders of magnitude: 15(S)-HpETE and 15(S)-HpEPE were the fastest and H_2_O_2_ was the slowest (Table 3). 

These reactivities explain the rapid inactivation of *Hs*Prx3 exposed to 20 µM _f_FA-OOH but not to 20 µM H_2_O_2_ as observed in the coupled catalytic assay (Figure 1). Indeed, k_hyperoxidation_ × [ROOH] is 520 s^−1^ for 15(S)-HpEPE_,_ 340 s^−1^ for 15(S)-HpETE, 6.2 s^−1^ for PGG_2_ and 0.022 s^−1^ for H_2_O_2_. Since hyperoxidation competes with resolution (2 s^−1^) for the sulfenic acid in Cys_P_ of the oxidized enzyme, expected amounts of hyperoxidized enzyme can be calculated as ~99% for 15(S)-HpEPE_,_ and 15(S)-HpETE, a fraction that decreased to ~70% for PGG_2_ and ~1% for H_2_O_2_ under the conditions employed in Figure 4. Indeed, *Hs*Prx3 hyperoxidation was detected at much lower concentrations of 15(S)-HpEPE_,_ and 15(S)-HpETE than of H_2_O_2_ and even at equimolar concentrations hyperoxidation at both Cys_P_ of the dimeric functional unit of *Hs*Prx3 is detected (as hyperoxidized monomers) (Figure 4). However, there is still a substantial proportion of the enzyme with one Cys_P_ bound to Cys_R_ (as hyperoxidized dimers) indicating that Cys_P_ of *Hs*Prx3 was not ~99% hyperoxidized. This probably reflects active site asymmetry in *Hs*Prx3, a dodecameric enzyme under reduced state [75], as already proposed for this and other members of the Prx family [37,58,74]. In this respect, it was previously reported that only half *Hs*Prx3 active sites were susceptible to hyperoxidation even at very high H_2_O_2_ concentrations (20 mM), forming hyperoxidized dimers as main products [76]. The slower hyperoxidation rate constant of *Hs*Prx3 by PGG_2_ with respect to 15(S)-HpEPE_,_ and 15(S)-HpETE is probably a consequence of its particular structure, since it is an endoperoxide-hydroperoxide, which could affect its interaction with *Hs*Prx3 active site.

Biophysical experiments using ANS showed a dose-dependent quenching of the intrinsic fluorescence of *Hs*Prx3 as well as a blue shift and an increase in ANS quantum yields in the presence of the protein (Figure 6A–C). These data suggest the presence of a hydrophobic region in *Hs*Prx3 accessible from the solvent and with nearby positive charges [61,62]. However, the Ka value obtained of 0.016 µM^−1^ is an order of magnitude lower than those reported for fatty acid binding proteins such as bovine serum albumin (0.32 µM^−1^) or even for the 1-Cys Prx from *Mycobacterium tuberculosis* AhpE (0.16 µM^−1^). Indeed, the Ka value for ANS binding to *Hs*Prx3 is in the range of those attributed to the binding of ANS to molten globule-like states of proteins [77]. Therefore, the existence of a well-defined binding site for hydrophobic compounds such as FA was not unambiguously proven by this technique. The ability of *Hs*Prx3 to bind fatty acids was further analyzed by measuring the effects of AA on the melting temperature of reduced *Hs*Prx3 (Figure 6D) and a model of interaction is proposed based on the results obtained using docking simulations (Figure 7). Indeed, in an inspection of the *Hs*Prx3 dimer structure and dynamics, we found and characterized a hydrophobic patch in close proximity to the enzyme active site that can allocate _f_FA-OOHs in such a way that may assist on their reduction by Cys_P_. It should be taken into account that the binding of _f_FA-OOH to *Hs*Prx3 is not of high affinity, since the non-oxidable fatty acid palmitic acid did not compete with ANS for the binding to *Hs*Prx3. The effect of low-affinity binding of non-reactive parts of substrates in catalysis was initially described by Jenks and coworkers [78] and is nowadays ascribed at least partially to changes in activation entropy of the reaction [79]. That was indeed the case in the one-cysteine Prx from *Mycobacterium tuberculosis* AhpE, that reacted much faster with _f_FA-OOH than with H_2_O_2_ at the expense of an important increase in the activation entropy of the Cys_P_ oxidation reaction [41].

From the kinetics of _f_FA-OOHs reduction by different mitochondrial peroxidases and other potential mitochondrial targets previously and herein reported (Table 4), it emerges that *Hs*Prx3 should be considered a preferential target for these species, particularly of 15(S)-HpETE and 15(S)-HpEPE. This is because of the fast rate constants of reaction combined with the high abundance of the enzyme in many cellular types, with reported concentration values in the 10–120 µM range [13]. In turn, GPx1 is oxidized by FA-OOHs with a similar rate constant (4 × 10^7^ M^−1^s^−1^ [28]) but its mitochondrial concentration is lower, so its contribution to _f_FA-OOH reduction is expected to be minor. Data on the kinetics of oxidation of *Hs*Prx5 by _f_FA-OOH are lacking so far, but if the high reactivity of the enzyme with other organic hydroperoxides is maintained (~10^7^ M^−1^s^−1^) [80], then Prx5 would also represent a main target of _f_FA-OOH since its mitochondrial concentration can attain 16 µM [81]. Of course, these calculations rely on the concentrations of the different peroxidases under reduced state, which can decrease if reduction by the mitochondrial reduction systems rate limits catalysis, or if the fraction of inactive enzymes (which includes but is not limited to hyperoxidized protein) increases. Similar calculations cannot be performed for PGG_2_ or similar endoperoxide-hydroperoxides since the kinetic data on its reactions with other mitochondrial targets are still lacking. 

Previous data indicated a mitochondrial formation flux of O_2_^•−^ of 0.8 µM/s in endothelial cells under normoglycemia, which can increase almost 10-fold under hyperglycemic conditions [82]. Most of it dismutates to H_2_O_2_ and O_2_ in a reaction catalyzed by Mn-superoxide dismutase (MnSOD), a highly efficient enzyme abundant in the organelle. Other potential targets of O_2_^•−^ include ^•^NO, which depending on its steady-state concentrations, can compete for a fraction of O_2_^•−^ and form peroxynitrite [4]. In sake of simplicity, we did not include in our simulations formation fluxes of ^•^NO which largely vary depending on conditions and cell type [4]. Considering mitochondrial formation fluxes of H_2_O_2_ in the 0.4–4 µM/s range and rate constants and concentrations indicated in Table 4, we determined expected yields of hyperoxidized Prx3 in the mitochondrial matrix as indicated in Table 5. Formation rates of _L_FA-OOH in mitochondrial membranes have been previously estimated as 0.1 µM/s [83]. Estimation of formation rates of _f_FA-OOH in the mitochondrial matrix are lacking, although they have been reported to largely vary under different physiopathological conditions [6]. So, we spanned a three order of magnitude range of _f_FA-OOH formation rates to investigate the expected effects on *Hs*Prx3 hyperoxidation. Time course simulations that were one hour long were performed in order to avoid other variables such as hyperoxidized *Hs*Prx3 sulfinic acid reduction by sulfiredoxin (which requires the translocation of sulfiredoxin, a cytosolic protein, to the mitochondria [84]) as well as protein turnover [85]. Due to the location of cytochrome *c* preferentially associated to the outer surface of the inner mitochondrial membrane, reduction of H_2_O_2_ (4.6 × 10^1^ M^−1^s^−1^) or _f_FA-OOH (k ~ 10^4^ M^−1^s^−1^) by cytochrome *c*/cardiolipin complexes [86] were not considered in the simulations related to *Hs*Prx3 hyperoxidation in the mitochondrial matrix shown in Table 5. However, with a calculated t_1/2_ value in the ms range (t_1/2_ = ln 2/∑ k × [peroxidases], see Table 4), the diffusion of a fraction of _f_FA-OOH formed in the mitochondrial matrix through the inner mitochondrial membrane to reach the cytochrome c/cardiolipin complex, a process that has been reported to be facilitated by particular proteins, is in principle possible. This would beparticularly important under conditions of increased oxidative stress, where the fraction of oxidized and hyperoxidized *Hs*Prx3 increases [87,88].

**Table 4 antioxidants-12-00408-t004:** Rate constants and concentrations used in simulations of expected yields of *Hs*Prx3 hyperoxidation.

Reaction	*k* (M^−1^s^−1^ or s^−1^)	Reference
(1) _red_Prx3 + H_2_O_2_ → Prx3SOH + H_2_O	2 × 10^7^	[12]
(2) Prx3SOH → Prx3SS	2	[13]
(3) Prx3SOH + H_2_O_2_ → Prx3SO_2_H + H_2_O	1.1 × 10^3^	[13]
(4) _red_Prx3 + _f_FA-OOH → Prx3SOH + _f_FA-OH	3.5 × 10^7^	This work
(5) Prx3SOH + _f_FA-OOH → Prx3SO_2_H + _f_FA-OH	1 × 10^7^	This work
(6) Prx3SS + _red_Trx2/Grx2 → _red_Prx3 + _ox_Trx2/Grx2	4 × 10^4 a^	[54]
(7) _red_Prx5 + H_2_O_2_ → Prx5SOH + H_2_O	3 × 10^5^	[80]
(8) _red_Prx5 + _f_FA-OOH → Prx5SOH + _f_FA-OH	2 × 10^7 b^	[80]
(9) Prx5SOH → Prx5SS	15	[80]
(10) Prx5SS + _red_Trx2 → _red_Prx5 + _red_Trx2	2 × 10^6^	[80]
(11) _red_GPx1 ^c^ + H_2_O_2_ → _ox_GPx1 + H_2_O	4 × 10^7^	[28]
(12) _red_GPx1 + _f_FA-OOH → _ox_GPx1 + _f_FA-OH	4 ×10^7^	[28]
(13) _ox_GPx1 + 2GSH ^d^→ _red_GPx1+ GSSG	2 × 10^5^	[28]

^a^ The rate constant of Prx3 disulfide (Prx3SS) reduction by the physiological reducing substrates Trx2 and/or Grx2 was set as 4 × 10^4^ M^−1^s^−1^ as estimated from the catalytic efficiency (k_cat_/Km) values reported in [54]. Trx2 plus Grx2 concentration in mitochondria was assumed to sum 10 µM [12] and was fixed under reduced state in the simulations (reduction of the redoxins is not rate limiting peroxide consumption). The concentration of Trx_2_ and Grx_2_ varies with the cellular type [54]. ^b^ The kinetics of the reactions of reduced *Hs*Prx5 with FA-OOHs is unknown so far. In these simulations, we assumed a rate constant similar to those of the enzyme oxidation by the non-natural organic hydroperoxides *tert*-butyl hydroperoxide and cumene hydroperoxides reported in [80]. The concentration of Prx5 was set as 16 µM as reported in [89]. The susceptibility of *Hs*Prx5 to hyperoxidation was reported to be low, so we did not include it in the simulations [90]. ^c^ Mitochondria express GPx1 with a concentration ~2 µM [91]. Although GPx4 can also be expressed in the mitochondria, particularly in testis, its expression levels are low in most somatic tissues and therefore it was not included in the simulations [30]. ^d^ Glutathione concentration was set at 5 mM and was fixed under reduced state, i.e., in these simulations it is assumed that reduction of glutathione by glutathione reductase/NADPH is faster than glutathione oxidation.

Under low physiological mitochondrial fluxes of H_2_O_2_ of ~0.4 µM/s, the amount of hyperoxidized *Hs*Prx3 formed in the mitochondrial matrix in one hour is ≤1 nM, unless a concomitant flux of _f_FA-OOH in the order of 0.001–0.1 µM/s occurs. On the contrary, at higher fluxes of H_2_O_2_ formation such as 4 µM/s, hyperoxidized enzyme can reach ~10–100 nM levels in the absence of fluxes of _f_FA-OOH, but still the latter are required for a significant fraction (≥1%) of the enzyme to be hyperoxidized in an hour. For example, for 2% hyperoxidation of the enzyme in an hour, a concomitant flux of 0.1 µM/s of _f_FA-OOH is needed in mitochondria expressing 100 µM *Hs*Prx3. In mitochondria expressing 20 µM *Hs*Prx3 formation rates of _f_FA-OOH required for a similar percentage of *Hs*Prx3 inactivation are in the nM/s range (Table 5). Thus, our results indicate that mitochondrial _f_FA-OOH contributes to the inactivation through hyperoxidation of *Hs*Prx3, a modification that has been detected in cells under both physiological and pathological conditions [49,50,92,93,94]. 

Similar data regarding the oxidation and hyperoxidation of the cytosolic Prx1 and Prx2 by _f_FA-OOH are lacking so far, with the only exception of the initial report by Cha et al. [34]. The peroxidatic active site in the three Prxs is highly conserved. More interestingly, all the residues that make up the hydrophobic patch in *Hs*Prx3 (Figure 7) are conserved in Prx2 (Appendix A). Furthermore, the reported rate constants of oxidation and hyperoxidation by H_2_O_2_ are similar [13,59,95,96]. Thus, it is tempting to speculate that the rate constants of oxidation and hyperoxidation by FA-OOH would probably be also similar, differences in susceptibility to hyperoxidation arising from differences in rates constants of resolution as in the case of H_2_O_2_. If that is the case, and since Prx2 is the one with the lower resolution rate constant, it would be expected to be the enzyme more prone to hyperoxidation also by _f_FA-OOH among the three. This is very interesting due to the different cellular compartmentalization of Prx2 (mostly cytosolic) with respect to Prx3 and due to the fact that depending on cells and conditions, _f_FA-OOH can be preferentially formed in different cell compartments. However, future work is required to test this hypothesis. 

Prxs and particularly, typical 2-Cys Prxs participate in redox signaling actions through different mechanisms. In the flood gate mechanism of redox signaling, hyperoxidation and the consequent inactivation of Prxs would allow the oxidation of other, less reactive hydroperoxide targets [97]. In the redox relay mechanism, Prxs act as sensors of hydroperoxides transferring the oxidation to signaling proteins [98,99]. We propose that *Hs*Prx3 oxidation and hyperoxidation by _f_FA-OOH reported herein contribute to both signaling mechanisms, either by increasing enzyme hyperoxidation yields but also serving as sensors of _f_FA-OOH in mitochondria. 

**Table 5 antioxidants-12-00408-t005:** Expected percentages of hyperoxidated *Hs*Prx3 in mitochondria forming different fluxes of H_2_O_2_ and _f_FAOOH.

% of Prx3-SO_2_^−^ with Respect to Initial Reduced Prx3				
**[Reduced Prx3] 20 µM ^a^**				
**_f_FA-OOH (µM/s)**	**H_2_O_2_ (µM/s)**	0.4	1	4
0		2.8 × 10^−3^	2 × 10^−2^	0.37
0.001		3.3 × 10^−2^	0.1	0.75
0.01		0.30	0.8	4
0.1		3.7	8.5	39
**[Reduced Prx3] 100 µM ^a^**				
**_f_FA-OOH (µM/s)**	**H_2_O_2_ (µM/s)**	0.4	1	4
0		1.5 × 10^−4^	9.3 × 10^−4^	1.6 × 10^−2^
0.001		2.0 × 10^−3^	5.6 × 10^−3^	3.6 × 10^−2^
0.01		1.9 × 10^−2^	4.8 × 10^−2^	0.22
0.1		0.23	0.51	2.1

^a^ Two different concentrations of Prx3 were included in the simulations, which are in the lower and higher range of those reported in the literature [12,14,100,101].

## 5. Conclusions

Herein, we demonstrate that *Hs*Prx3 is rapidly oxidized and hyperoxidized by _f_FA-OOH, with rate constants that are in the order of ≥3.5 × 10^7^ and ~10^7^ M^−1^s^−1^ for the AA-derived 15(S)-HpETE and EPA-derived 15(S)-HpEPE, respectively. Compared with other biologically relevant hydroperoxides such as H_2_O_2_ and peroxynitrite, the rate constants of *Hs*Prx3 oxidation by these _f_FA-OOHs are in the same order of magnitude, but hyperoxidation rate constants are considerably higher (Table 3). The endoperoxide-hydroperoxide PGG_2_, also derived from AA, oxidized *Hs*Prx3 almost as rapidly but showed a lower rate constant of hyperoxidation. We present a model of interaction of _f_FA-OOH with *Hs*Prx3 in which the former binds to a hydrophobic patch of the enzyme and positions the hydroperoxide group in close proximity to Cys_P_ that would facilitate the reaction. We propose that, as for other mitochondrial hydroperoxides such as H_2_O_2_ and peroxynitrite, *Hs*Prx3 is expected to be a main target for _f_FA-OOH at least under low oxidative stress conditions in which most of the enzyme is under reduced state. Finally, our results indicate that low fluxes of _f_FA-OOHs (in the nM/s range) formed in the mitochondrial matrix are expected to increase hyperoxidation yields of *Hs*Prx3 in mitochondria under different conditions. Considering the key roles of Prxs in redox signaling processes we propose *Hs*Prx3 not only as a reductant but also as a sensor of _f_FA-OOHs formed in mitochondria.

## Figures and Tables

**Figure 1 antioxidants-12-00408-f001:**
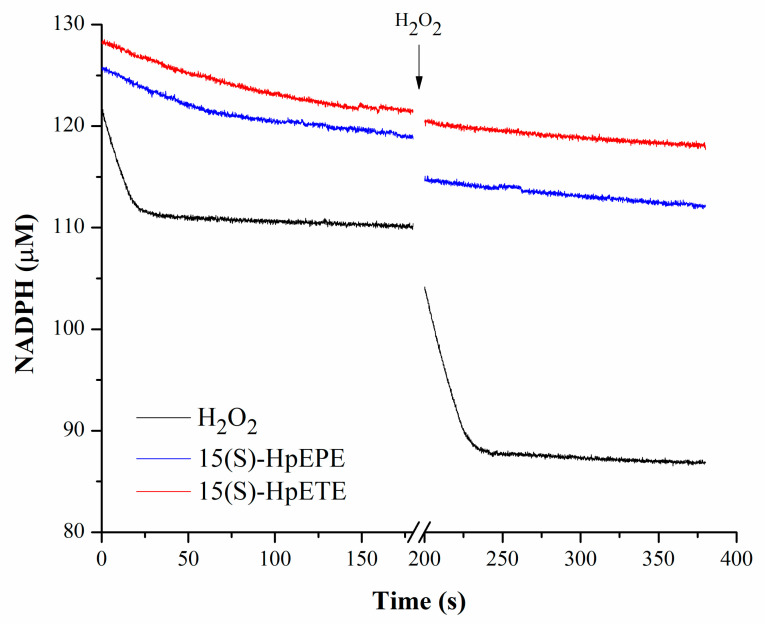
Inactivation of *Hs*Prx3 by _f_FA-OOHs. NADPH was pre-incubated with *Eg*TR (40 nM), *Hs*Trx1 (30 µM) and *Hs*Prx3 (0.8 µM) and rapidly mixed with the indicated oxidizing substrates (20 µM) (black, H_2_O_2_; blue, 15(S)-HpEPE; red, 15(S)-HpETE) in sodium phosphate buffer 50 mM plus 0.1 mM DTPA, pH 7.8 and 25 °C. The arrow indicates the addition of H_2_O_2_ (20 µM) to the mixture.

**Figure 2 antioxidants-12-00408-f002:**
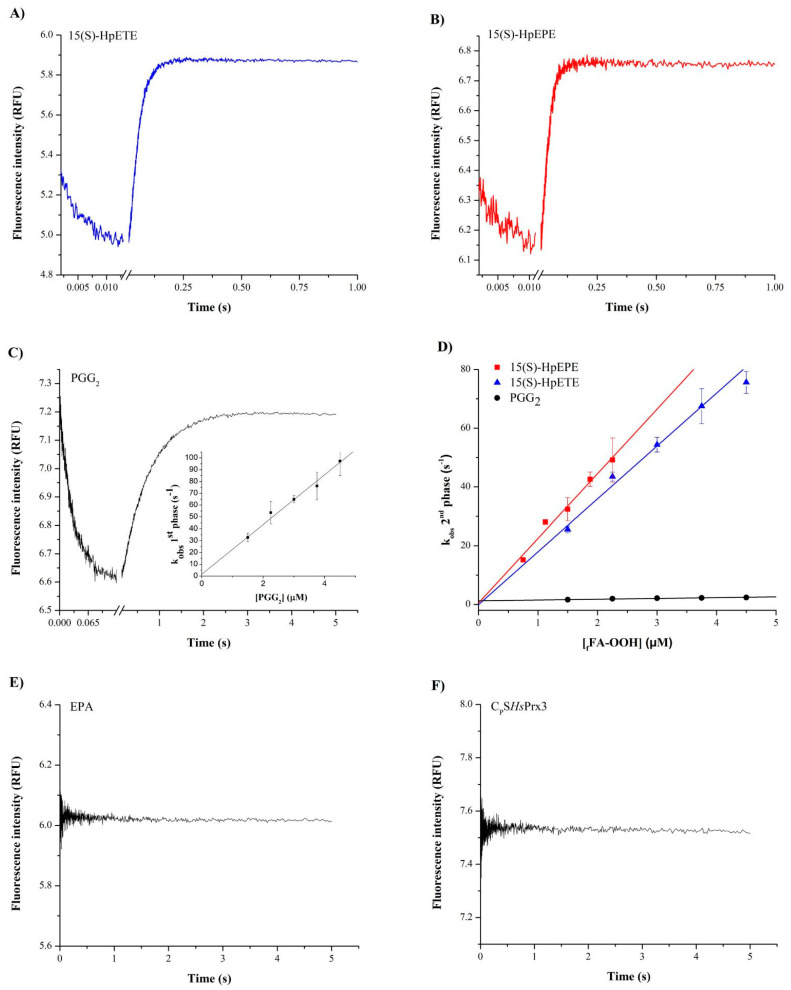
Kinetics of *Hs*Prx3 oxidation and hyperoxidation by _f_FA-OOHs. Reduced *Hs*Prx3 (0.3 µM final concentration) was rapidly mixed with 15(S)-HpETE (**A**), 15(S)-HpEPE (**B**) and PGG_2_ (**C**) (1.5 µM final concentration) in sodium phosphate buffer 50 mM plus 0.1 mM DTPA at pH 7.8 and 12 °C. The time courses of intrinsic fluorescence intensity change were followed (λ_ex_ = 295 nm, total emission). The inset of (**C**) shows a linear fit of *k*_obs_ of fluorescence intensity decay plotted versus PGG_2_ concentration from which the second order rate constant of *Hs*Prx3 oxidation by PGG_2_ was obtained. The rate constant of hyperoxidation of Cys_P_ was determined for the three _f_FA-OOHs mentioned above, plotting the *k*_obs_ of the phases of fluorescence intensity recovery against _f_FA-OOH concentration (**D**). Both in inset of (**C**) and in (**D**) the symbols and error bars represent medians and standard deviations from at least 5 replicates of one of the three independent experiments performed. The effect of EPA (3 µM) on the intrinsic fluorescence of reduced *Hs*Prx3 (0.3 µM) (**E**) and of PGG_2_ (0.3 µM) on C_P_S*Hs*Prx3 (0.3 µM) (**F**) are also shown.

**Figure 3 antioxidants-12-00408-f003:**
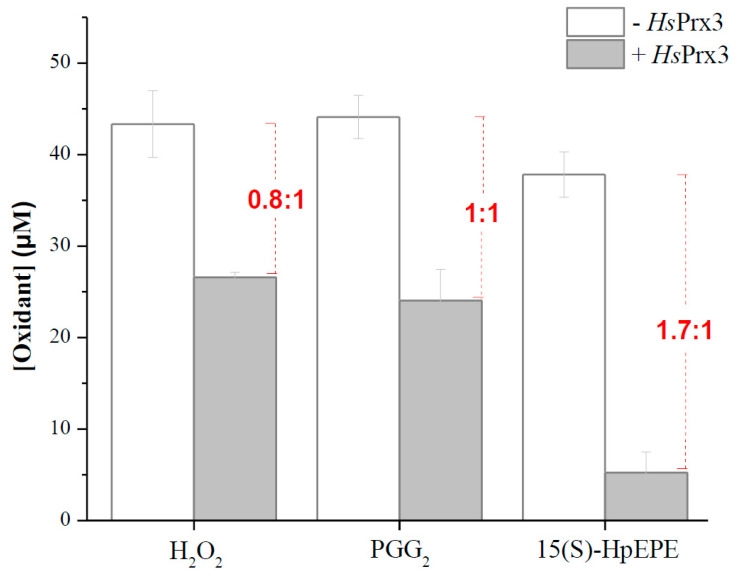
Stoichiometry of hydroperoxide consumption by *Hs*Prx3. 15(S)-HpEPE and PGG_2_ consumption by *Hs*Prx3. H_2_O_2_, PGG_2_ and 15(S)-HpEPE (left to right) decay in sodium phosphate buffer 50 mM (pH 7.8, 25 °C) in the absence (white bars) or 30 s after the addition of 20 µM reduced *Hs*Prx3 (grey bars). Numbers in red represent the stoichiometry of hydroperoxide consumption.

**Figure 4 antioxidants-12-00408-f004:**
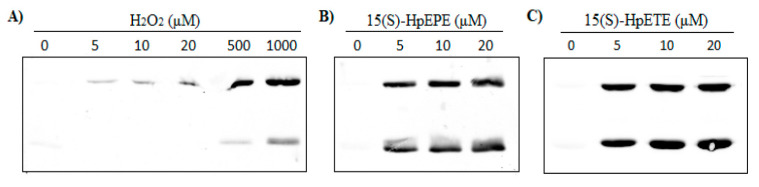
Hyperoxidation of *Hs*Prx3 by different hydroperoxides. Reduced *Hs*Prx3 (5 µM) was treated with the indicated concentrations of (**A**) H_2_O_2_; (**B**) 15(S)-HpEPE; and (**C**) 15(S)-HpETE in sodium phosphate buffer 50 mM pH 7.8 plus 0.1 mM DTPA. The samples were alkylated and used for Western Blot detection of Cys_P_ hyperoxidation.

**Figure 5 antioxidants-12-00408-f005:**
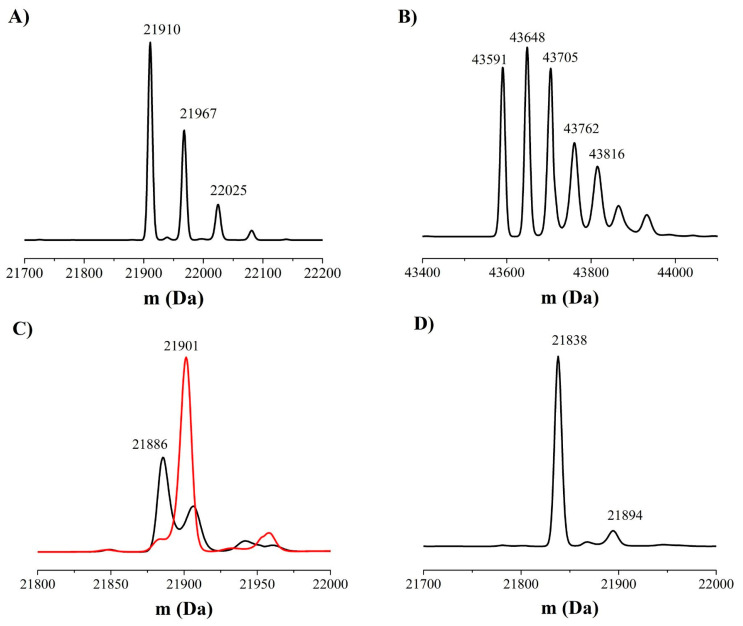
MS analysis of the modifications of *Hs*Prx3 caused by 15(S)-HpEPE. Mass spectrum of (**A**) reduced *Hs*Prx3 (10 µM); (**B**) reduced *Hs*Prx3 treated with H_2_O_2_ (20 µM); (**C**) reduced *Hs*Prx3 treated with 15(S)-HpEPE (10 µM, black trace; and 20 µM, red trace); and (**D**) reduced C_P_S*Hs*Prx3 treated with 15(S)-HpEPE (20 µM). All samples were alkylated with excess 2-iodoacetamide after treatment (5 mM).

**Figure 6 antioxidants-12-00408-f006:**
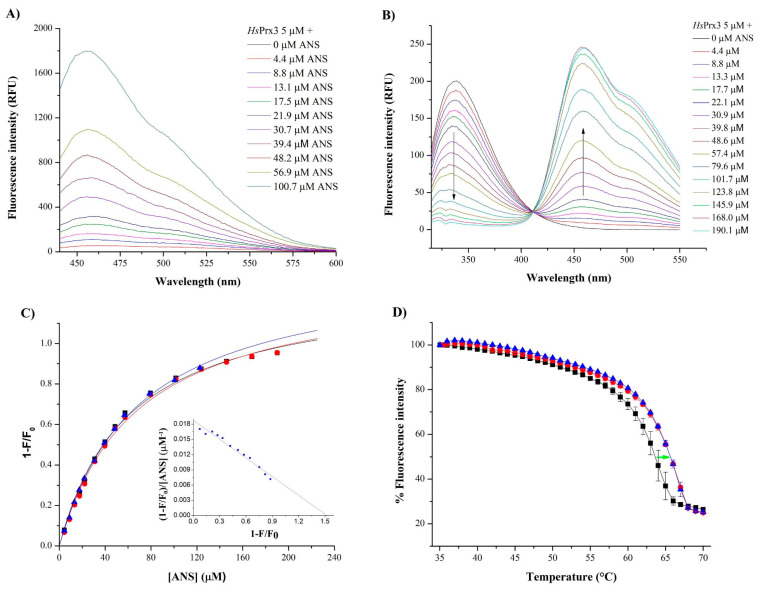
Binding of ANS and AA to *Hs*Prx3. (**A**) Direct fluorescence measurement of ANS at an excitation wavelength of 375 nm. For each ANS concentration, the spectrum shown corresponds to the difference between the recorded spectra in the presence and in the absence of *Hs*Prx3 (5 µM). (**B**) *Hs*Prx3 (5 µM) was incubated with increasing ANS concentrations and fluorescence emission spectra (λ_ex_ = 295 nm) were registered. The arrows indicate the direction of fluorescence changes observed after addition of different concentrations of ANS. (**C**) For three different protein concentrations (2 (▲), 5 (●) and 8 (■) µM), relative intrinsic fluorescence changes were plotted against [ANS], resulting in saturation curves which were fitted to hyperbolic functions. An example of the linearization through Scatchard plot corresponding to 2 µM *Hs*Prx3 is shown in the inset. (**D**) Heat unfolding transitions following the intrinsic fluorescence intensity (λ_ex_ = 295 nm, λ_em_ = 335 nm) of the tryptophans of *Hs*Prx3 (4 μM) in the absence (■) and presence of AA (10 μM (●) and 20 μM (▲)) and DTT (2 mM). Green arrow shows the shift in the apparent Tm in the presence of AA.

**Figure 7 antioxidants-12-00408-f007:**
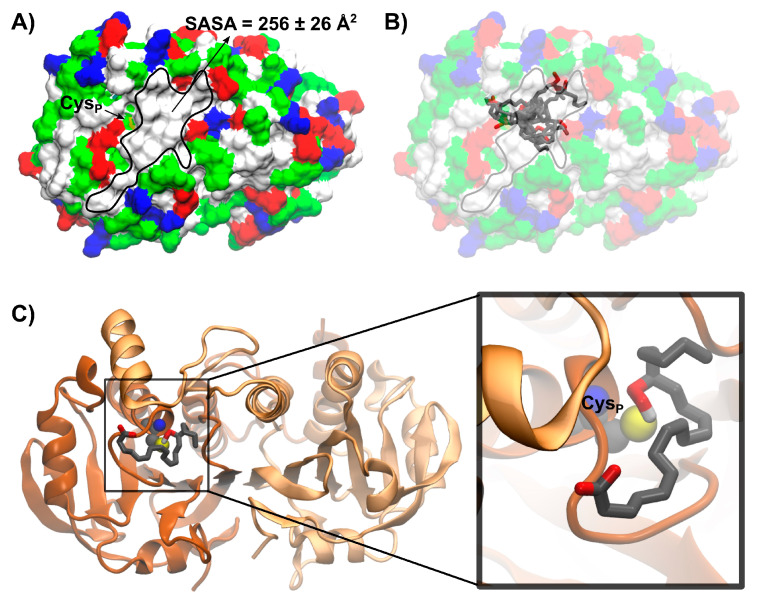
Best ranked structures of the binding of 15(S)-HpETE to *Hs*Prx3 obtained by docking simulations. (**A**) Surface representation of *Hs*Prx3 dimer colored by amino acid properties: white, hydrophobic; green, polar; blue, basic; red, acidic. The hydrophobic patch close to Cys_P_ is highlighted and the solvent accessible surface area (SASA) value measured during the MD simulation is shown. (**B**) Superimposed 15(S)-HpETE representative structures of the ten best ranked clusters with at least five structures obtained by molecular docking. Hydrogen atoms are not shown for simplicity. (**C**) Representative structure of the most populated best ranked cluster from the molecular docking of 15(S)-HpETE to the *Hs*Prx3 dimer structure. Only the hydroperoxide hydrogen atom is shown for simplicity.

**Table 1 antioxidants-12-00408-t001:** Catalytic activity of *Hs*Prx3 using different oxidizing substrates.

Oxidizing Substrate	v_o_ ^1^ (µM/s)	v_o_ after Subsequent Addition of 20 µM H_2_O_2_ (µM/s)
H_2_O_2_	0.55	0.60
15(S)-HpETE	0.07	0.03
15(S)-HpEPE	0.08	0.02
PGG_2_	0.09	0.08

^1^ Initial rate here means the rate at the beginning of data acquisition, 15 s after the addition of the oxidizing substrate.

**Table 2 antioxidants-12-00408-t002:** Mass spectrometry analysis of *Hs*Prx3 modifications caused by H_2_O_2_ and 15(S)-HpEPE.

Treatment	Mass (Da)	Peak Description
**Reduced *Hs*Prx3**	21,910	*Hs*Prx3-2CAM
	21,967	*Hs*Prx3-3CAM
	22,025	*Hs*Prx3-4CAM
***Hs*Prx3 + H_2_O_2_**	43,591	*Hs*Prx3 dimer
	43,648	*Hs*Prx3-CAM dimer
	43,705	*Hs*Prx3-2CAM dimer
	43,762	*Hs*Prx3-3CAM dimer
	43,816	*Hs*Prx3-4CAM dimer
***Hs*Prx3 + 15(S)-HpEPE**	21,886	CAM-*Hs*Prx3-SO_2_
	43,696	CAM-*Hs*Prx3-SO_3_ dimer
	21,901	CAM-*Hs*Prx3-SO_3_
**C_P_S*Hs*Prx3 * + 15(S)-HpEPE**	21,838	C_P_S*Hs*Prx3-CAM
	21,894	C_P_S*Hs*Prx3-2CAM

* the predicted molecular weight of wild type and C_P_S*Hs*Prx3 reduced subunit are 21,797 Da [13] and 21,781 Da, respectively.

**Table 3 antioxidants-12-00408-t003:** Kinetics of *Hs*Prx3 oxidation and hyperoxidation by biologically relevant hydroperoxides.

Hydroperoxide	k_ox._ (M^−1^s^−1^)	k_hyperox._ (M^−1^s^−1^)	Reference
H_2_O_2_	2.0 × 10^7 (1)^	6 × 10^3 (1)^; 1.1 × 10^3 (2)^	[12,13,15]
Peroxynitrite	1.0 × 10^7 (3)^	Not determined	[13]
15(S)-HpEPE	˃3.5 × 10^7 (4)^	(2.6 ± 0.4) × 10^7 (4)^	This work
15(S)-HpETE	˃3.5 × 10^7 (4)^	(1.7 ± 0.1) × 10^7 (4)^	This work
PGG_2_	2.4 × 10^7 (4)^	(3.1 ± 0.7) × 10^5 (4)^	This work

^(1)^ pH 7.4, 25 °C; ^(2)^ pH 7.8, 14 °C; ^(3)^ pH 7.8, 25 °C; ^(4)^ pH 7.8, 11–12 °C. Mean values and standard deviations of the rate constants were calculated from three independent experiments.

## Data Availability

The data are contained within this article.

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
