# Peer review of "Mitochondrial Peroxiredoxin 3 Is Rapidly Oxidized and Hyperoxidized by Fatty Acid Hydroperoxides"

_antioxidants, 2023, doi:10.3390/antiox12020408_

Round 1

Reviewer 1 Report

The manuscript "Mitochondrial peroxiredoxin 3 is rapidly oxidized and hyper-oxidized by fatty acid hydroperoxides" by Cardozo et al reports their investigation into the peroxidase activity of the human Pxdr3 towards oxidant species such as free fatty acid hydroperoxides (fFA-OOHs ). Like other forms of hydroperoxides such as H202 and peroxynitrite, the FA-OOHs participate in cellular redox signaling but at over concentration can lead to mitochondrial dysfunction, cell cytotoxicity and diseases states.

Prx3 catalyzes the reduction of most of mitochondrial H2O2 and as previously indicated by the authors, Prx3 is also an efficient peroxynitrite reductase.

In this work, the authors proposed HsPrx3 not only as a reductant but also as a sensor of fFA-OOHs formed in mitochondria.

By using different biochemical and biophysical methodologies, Cardoso et al  clearly demonstrated that Prx3 is able to catalyze in vitro the reduction of fFA-OOHs. Prdx3 is oxidized and hyperoxidized by fFA-OOHs. Compared with H2O2 and peroxynitrite, the rate constants of Prx3 oxidation by these fFA-OOHs showed to be in the same order of magnitude, but hyperoxidation rate constants were considerably higher.

The authors also suggested a model of interaction of fFA-OOH with HsPrx3 in which the former binds to a hydrophobic structures of the enzyme in close proximity to CysP that would facilitate the reaction. Kinetic simulation analysis supported that mitochondrial fFA- OOHs formation fluxes in the range of nM/s are expected to contribute to HsPrx3 hyperoxidation, a modification that has been detected in vivo under physiological and pathological conditions.

The whole experimental design and the clarity of all procedures and results are very good.   

In summary the results of this manuscript seemed to be relevant for the field and in particular for the readers of the Antioxidant journal.

minor

Lane  76-77 “Cha et al in 2000 [34] that reported that human Prx24 rapidly reduces FA-OOH and interacts with erythrocyte plasma membrane through a its C-terminal region” 

Author Response

We thank reviewer 1 for his appreciation of our work. The minor correction indicated was done (in fact we substituted “that” by “who”). It is now indicated in the manuscript

“The ability of some members of the Prx family to interact with phospholipid membranes and to reduce FA-OOHs was firstly demonstrated by Cha et al in 2000 [34] who reported that human Prx2[1] rapidly reduces FA-OOH and interacts with erythrocyte plasma membrane through a C-terminal region [34]”

[1] Named thiol-specific antioxidant protein 1 in [34]

Reviewer 2 Report

Review of Cardozo et al – Antioxidants – 2023

In this paper, Cardozo et al. analyze the reaction of human mitochondrial Prx3 with fatty hydroperoxides as substrates and compared it to the well-known H2O2 substrate. This is an important matter as Prx3 is a potential target of fatty acid hydroperoxides released from oxidized membranes in mitochondria. The experimental strategy includes kinetic analyses of the peroxidase and hyperoxidation reactions, mass spectrometry analyses of the oxidation products, and complementary biophysical analyses and molecular modeling to assess potential binding of the hydrophobic substrates to Prx3. A main asset of the study is exploitation of the measured kinetic constants to simulate HsPrx3 hyperoxidation in vivo, supporting the role of this enzyme in fatty hydroperoxides metabolism.

Main points

- The mechanism of hyperoxidation of Prxs from the AhpC/Prx1 and other Prx classes have been established to be regulated by the competition of the sulfenic acid intermediate (in a fully folded conformation) between the sulfinylation reaction and conformational transition to a locally unfolded LU form, leading next to the resolution reaction. Thus, the resolution event (disulfide formation) is not in direct competition with hyperoxidation, and this likely applies to most Prx1 family enzymes (Kriznik et al, ACS Catal. 2020, 10, 3326−3339). This point should be corrected or modulated in the text, e.g. lines 44, 170, 268, 475.

- Figure 2: Trp-fluorescence based rapid kinetic analyses allows to measure the rate constant of the conformational transition, corresponding to the y-axis intercept of the second increasing phase kobs plot against peroxide concentration, as observed in a previous paper of this group (Free Radical Biology and Medicine 130 (2019) 369–378). In the present study, Fig 2D shows intercepts at 0 s-1, while for HsPrx3 this conformational transition was measured between 2 and 20 s-1. This is surprising and is not in accordance with the competition mechanism. Could the authors explain this point in the text?

- Furthermore, these very high hyperoxidation values raise the question of the molecular bases of this huge reactivity. Although docking results suggest that there could be binding, the distance between the peroxide group and Cys CP thiolate (4.5 Å) does not reflect a productive complex. This this point should be further discussed in the manuscript.

- Figure 6: Substrate recognition in the active site is proposed based on ANS and thermostability experiments. Since ANS seems to bind in a defined site (stoichiometry close to 1), could the ANS signal be used to measure the binding of fatty hydroperoxide/acids to HsPrx3 by competition?

- statistical analysis and replicates information are missing all along the manuscript, please provide this information.

Minor points

Table 1: please report the results as rate constants and not rates

line 151, NAPDH concentration is missing

line 152, in the coupled peroxidase assay did you include basal activity measurement to account for non-specific Trx system oxidation by peroxide?

line 174, please rephrase “rapidly mixed….stoppedflow spectrofluorimeter”. Your sentence let the reader think that Prx and peroxides were premixed before the measure in the stopped-flow apparatus.

line 183, did you include a precipitation step before alkylation?

line 264, please show data on Figure 1

line 288: specify that “oxidation” corresponds to the sulfenic acid formation

line 317, the figure 3 is not cited in the 3.3 paragraph.

line 321, Does “Moiety” means “equivalent” in the sentence “the stoichiometry of hydroperoxide consumption 3 was higher (1.7:1) for 15(S)-HpEPE. This is consistent with a fast hyperoxidation process in the case of 15(S)-HpEPE that consumes a second moiety of oxidant.” Please clarify

line 322, there are error bars but no information about original data nor replicates.

line 351, could you provide data for PGG2?

Figures 2, 6, 7 : quality should be improved.

It would be helpful to add in supplemental section the chemical structures of the fatty hydroperoxides under study.

Line 445: “enzymatic”

Table 5: format issues in the third line (H2O2 µM/s), there is a shift

Supplementary Fig. 1, a piece of the right panel is missing.

Supplementary Fig. 2, improve quality.

Author Response

Responses to the reviewer 2

In this paper, Cardozo et al. analyze the reaction of human mitochondrial Prx3 with fatty hydroperoxides as substrates and compared it to the well-known H2O2 substrate. This is an important matter as Prx3 is a potential target of fatty acid hydroperoxides released from oxidized membranes in mitochondria. The experimental strategy includes kinetic analyses of the peroxidase and hyperoxidation reactions, mass spectrometry analyses of the oxidation products, and complementary biophysical analyses and molecular modeling to assess potential binding of the hydrophobic substrates to Prx3. A main asset of the study is exploitation of the measured kinetic constants to simulate HsPrx3 hyperoxidation in vivo, supporting the role of this enzyme in fatty hydroperoxides metabolism.

Main points

- The mechanism of hyperoxidation of Prxs from the AhpC/Prx1 and other Prx classes have been established to be regulated by the competition of the sulfenic acid intermediate (in a fully folded conformation) between the sulfinylation reaction and conformational transition to a locally unfolded LU form, leading next to the resolution reaction. Thus, the resolution event (disulfide formation) is not in direct competition with hyperoxidation, and this likely applies to most Prx1 family enzymes (Kriznik et al, ACS Catal. 2020, 10, 3326−3339). This point should be corrected or modulated in the text, e.g. lines 44, 170, 268, 475.

We thank the reviewer for noting this. We have indicated in the text of the revised version that resolution involves a change in conformation between a fully folded to locally unfolded conformation followed by the reaction between the sulfenic acid and the resolving cysteine of a second subunit to form a disulfide bond.

- Figure 2: Trp-fluorescence based rapid kinetic analyses allows to measure the rate constant of the conformational transition, corresponding to the y-axis intercept of the second increasing phase kobs plot against peroxide concentration, as observed in a previous paper of this group (Free Radical Biology and Medicine 130 (2019) 369–378). In the present study, Fig 2D shows intercepts at 0 s-1, while for HsPrx3 this conformational transition was measured between 2 and 20 s-1. This is surprising and is not in accordance with the competition mechanism. Could the authors explain this point in the text?

Using the change in fluorescence we have previously reported a k of conformational transition of 2 s-1 at pH 7.8 and 14oC (in the FRBM paper indicated by the reviewer), using hydrogen peroxide as oxidizing substrate. It is a bit lower (~1.5 s-1) at pH 7.8 and 11-12oC. The intercepts shown in Fig 2D is a bit variable, depending on the hydroperoxide, it goes from 0 - 2 s-1. We think that this variability depends on the fact that since the reaction of hyperoxidation is so fast, at the lower concentrations of hydroperoxides shown in the Fig 2D, real pseudo-first order conditions (10x with respect of the enzyme) did not apply, even when we worked at low concentration of the enzyme (0.3 µM). We could not work at lower concentration of the enzyme because the signal was too small, and we could not further increase the concentration of the hydroperoxide since the reaction was so fast that it occurred in the mixing time of the stopped flow apparatus. That made the offset (intercept) to have more noise than we would like.

- Furthermore, these very high hyperoxidation values raise the question of the molecular bases of this huge reactivity. Although docking results suggest that there could be binding, the distance between the peroxide group and Cys CP thiolate (4.5 Å) does not reflect a productive complex. This this point should be further discussed in the manuscript.

The discussion section was changed in order to introduce concepts that were not clear in the first version of the manuscript, for example the fact that the affinity of the non-reactive part of the fatty acid hydroperoxide to HsPrx3 is not very high, and that it has already been reported by Jenks et al in other enzymes this effect, which has been called the “Circe effect”. Please see our responses to the major points addressed by the second reviewer and the discussion section in the revised version of the manuscript.

- Figure 6: Substrate recognition in the active site is proposed based on ANS and thermostability experiments. Since ANS seems to bind in a defined site (stoichiometry close to 1), could the ANS signal be used to measure the binding of fatty hydroperoxide/acids to HsPrx3 by competition?

Yes, it could, and in fact we made the attempt, but at concentrations of fatty acid lower than CMC they did not compete with ANS, indicating that the affinity for the fatty acid is not very high. It is now included in the revised version of the manuscript. That is not contradictory with the increase in reactivity observed, since low affinity binding of substrates have been reported to have important effects on catalysis by different authors (example Jenks et al) as indicated in the revised version of the manuscript. Please see our responses to the second reviewer also.

- statistical analysis and replicates information are missing all along the manuscript, please provide this information.

We have included statistical analysis and replicates information in the revised version of the manuscript.

Minor points

Table 1: please report the results as rate constants and not rates

The rate constants and the corresponding standard deviations are given in Table 3 in the revised version of the manuscript.

line 151, NAPDH concentration is missing

Done

line 152, in the coupled peroxidase assay did you include basal activity measurement to account for non-specific Trx system oxidation by peroxide?

It was included.

line 174, please rephrase “rapidly mixed….stoppedflow spectrofluorimeter”. Your sentence let the reader think that Prx and peroxides were premixed before the measure in the stopped-flow apparatus.

It was done.

line 183, did you include a precipitation step after alkylation?

No, we did not include a precipitation step.

line 264, please show data on Figure 1

I am sorry we did not understand this comment.

line 288: specify that “oxidation” corresponds to the sulfenic acid formation

Done

line 317, the figure 3 is not cited in the 3.3 paragraph.

Done

line 321, Does “Moiety” means “equivalent” in the sentence “the stoichiometry of hydroperoxide consumption 3 was higher (1.7:1) for 15(S)-HpEPE. This is consistent with a fast hyperoxidation process in the case of 15(S)-HpEPE that consumes a second moiety of oxidant.” Please clarify

We changed moiety by equivalent as suggested in seek of clarity.

line 322, there are error bars but no information about original data nor replicates.

Number of replicates and independent experiments were included in the revised version of the manuscript

line 351, could you provide data for PGG2?

The addition of PGG2 to reduced Prx3 caused the formation of hyperoxidized dimers (dimers with one peroxidatic cysteine hyperoxidized and the other forming a disulfide bridge with the resolving Cys), like in the case of H2O2 at the same concentration, but not the formation of hyperoxidized monomers (both peroxidatic Cys hyperoxidized). In the case of HpETE at the same concentration, both hyperoxidized dimers and monomers are observed, which is consistent with the higher rate constant of hyperoxidation of HpETE compared with PGG2 and H2O2.

line 352, loading control of experiment? Wb wit hPrx3 ab ? Coomassie?

As indicated under methods, the SDS Page gel shown in Supplementary material is stained with Coomasie, while in the western shown in figure 4 an antisulfinic/sulfonic acid antibody was used.

Figure s2, 6, 7 : quality should be improved.

We agree with the reviewer, the Figures were changed by higher quality figures.

It would be helpful to add in supplemental section the chemical structures of the fatty hydroperoxides under study.

Done

Line 445: enzymatic

Done

Table 5: format issues in the third line (H2O2 µM/s), there is a shift

The format of Table 5 was corrected

Reviewer 3 Report

Review of antioxidants-2154269

This manuscript presents strong evidence, from multiple approaches, for a high reactivity of fatty acid hydroperoxides with reduced Prdx3, in line with that of the “canonical” substrates H2O2 and ONOOH, and a surprisingly high reactivity of fatty acid hydroperoxides with the Prdx3 sulfenic acid, which is much higher than that of H2O2. These are novel and relevant findings, and the work is sound to the best of my understanding. The manuscript is mostly clear and well written. I have just a couple of concerns about how the data are presented. The first is that the numbers of replicates and independent experiments are never presented, which does not allow to assess the reproducibility and dispersion of the results. The second related concern is that, in addition, no standard errors (or some other valid dispersion statistics) for the newly determined rate constants are presented in Table 3. This is essential to assess, for instance, if the khydroperox for 15(S)-HpEPE and 15(S)-HpETE are significantly different.

In the Discussion the authors may want to more explicitly discuss the reasons underlying the much higher reactivity of 15(S)-HpETE and 15(S)-HpEPE with HsPrdx3-SOH relative to H2O2 and other hydroperoxides. The lower reactivity of PGG2 with HsPrdx3-SOH suggests that specific binding to the hydrophobic patch may play a role in facilitating the reaction. However, a lower affinity of PGG2 for this binding site should also reflect on kox, which does not appear to be much lower for this hydroperoxide. So, could there be other factors into play?

Another important issue for the physiological relevance of the oxidation and hyperoxidation of HsPrdx3 by free-fatty-acid hydroperoxides is whether in vivo the likely more abundant non-peroxidized free fatty acids outcompete the hydroperoxides for the HsPrdx3 binding site, thus inhibiting the oxidation of the latter. Can the authors say anything about this issue based on their docking and MD simulations? Eg. does the -OOH group contribute significantly for binding energy?

 Line 416: Footnote 5 is hidden

MINOR ISSUES

Line 16: Please pay attention to superscripts in the revised version

Line 17: please write the meaning of the PGG2 abbreviation in the abstract.

Line 63: Please consider “Both, free and lipid-bound FA-OOHs…”

Line 150: Please consider “DTPA” and define the abbreviation.

Line 154: Readers may appreciate some explanation of why EgTR is preferable to other Thioredoxin Reductases for this assay.

Line 187: Please define abbreviation “ME” at first mention. Also, a betta appears to be missing, as per the abbreviations list.

Lines 276-281: Figure 1. Was this experiment repeated to check for reproducibility?
In this and subsequent figures, please pay attention to the layout of the legends, as the text in lines 278-281 (in this case) is formatted as the main text, which may confuse the typesetter.

Table 1: Where there replicates, repeats? How many?

Line 290: Consider “[…] the first phase was so rapid that it was complete in a few ms […]”

Figure 2: How many experiments and/or replicates were made? What do the dots and error bars in panels C, D represent? Are the dots means/medians from independent experiments or from replicates? Do the error bars represent standard deviations, standard errors of the mean, or something else?

Line 361: Consider “Treatment with excess 2-iodoacetamide caused alkylation of reduced HsPrx3 […]”

Lines 369-376: Don’t these observations suggest that oxidation of the sulfinic acid to a sulfonic acid by 15(S)-HpEPE is not much slower than oxidation of the sulfenic acid to the sulfinic acid?

Figure 5 and Table 2: Since HsPrdx3 without the targeting sequence has just three Cys residues, the observation of HsPrx3-4CAM needs some explanation.

Lines 527-530: Considering typical PLA2 and GPX4 activities in mitochondria, just a minor fraction of phospholipid hydroperoxides should be released as fFA-OOHs before the hydroperoxide being reduced via GPX4 [e.g. doi:10.1016/0891-5849(95)00040-5]. Thus, this seems to be an unlikely source of fFA-OOHs capable of significantly hyperoxidizing HsPrdx3 unless GPX4 is downregulated.

Line 551: Consider “reduction of the redoxins […]”

Table 5: The table is somewhat confusing as presented. First, it is mis-formatted, with the columns mis-aligned with the respective headers, which makes it difficult to read. Second, the title is inconsistent with the contents, since what is actually presented are the Prx3-SO2- concentrations, in µM, not the reaction yields, which are dimensionless quantities. And since the main take are the fractions of Prdx3 hyperoxidized in each condition, the message will perhaps be better conveyed by presenting these fractions (by dividing the [Prx3-SO2- ] by the total Prdx3 concentration in each case) instead of [Prx3-SO2- ].

Author Response

We also thank reviewer 3 for his comments and corrections that we consider will result in a better understanding of the manuscript.

This manuscript presents strong evidence, from multiple approaches, for a high reactivity of fatty acid hydroperoxides with reduced Prdx3, in line with that of the “canonical” substrates H2O2 and ONOOH, and a surprisingly high reactivity of fatty acid hydroperoxides with the Prdx3 sulfenic acid, which is much higher than that of H2O2. These are novel and relevant findings, and the work is sound to the best of my understanding. The manuscript is mostly clear and well written. I have just a couple of concerns about how the data are presented. The first is that the numbers of replicates and independent experiments are never presented, which does not allow to assess the reproducibility and dispersion of the results. The second related concern is that, in addition, no standard errors (or some other valid dispersion statistics) for the newly determined rate constants are presented in Table 3. This is essential to assess, for instance, if the khydroperox for 15(S)-HpEPE and 15(S)-HpETE are significantly different.

We thank the reviewer for his comments. We have now included the number of replicates in rate constants determinations shown in Figure 2, and also the number of independent experiments in Table 3, where we are now reporting not only the mean value but also the SD of the rate constants.

In the Discussion the authors may want to more explicitly discuss the reasons underlying the much higher reactivity of 15(S)-HpETE and 15(S)-HpEPE with HsPrdx3-SOH relative to H2O2 and other hydroperoxides. The lower reactivity of PGG2 with HsPrdx3-SOH suggests that specific binding to the hydrophobic patch may play a role in facilitating the reaction. However, a lower affinity of PGG2 for this binding site should also reflect on kox, which does not appear to be much lower for this hydroperoxide. So, could there be other factors into play?

This is a very important issue that we consider was not clear enough in the previous version of the manuscript and we made our best to make it clear in this revised version. Although we do see a binding site of fatty acid hydroperoxides in HsPrx3, we do not consider that it is a high affinity binding site. Indeed, the Ka for ANS was not very high, as already indicated in the manuscript, and fatty acid (at concentrations below CMC) were not capable of competing with ANS for HsPrx3 (we have added this information in the revised version of the manuscript). Indeed, a great effect of a non-high affinity binding of non-reactive parts of substrates on the reactivity of enzymes with those substrates has been repeatedly observed by different authors, including the seminal works of Jenks, and it is part of the so called “Circe effect”. Nowadays, part of the Circe effect in at least some enzymes is ascribed to effects in activation entropy of the reaction catalyzed (refs 83 and 84 in the revised manuscript). In the particular case of peroxiredoxins, we determined that an increase in activation entropy is the main responsible of the highest reactivity of the peroxidatic cysteine of the 1 Cys peroxiredoxin from Mycobacterium tuberculosis AhpE with fatty acid hydroperoxides with respect to hydrogen peroxide. The activation enthalpy of the reaction was almost the same for both substrates (ref 67 in the revised manuscript). We have changed the discussion section of the manuscript accordingly. 

Another important issue for the physiological relevance of the oxidation and hyperoxidation of HsPrdx3 by free-fatty-acid hydroperoxides is whether in vivo the likely more abundant non-peroxidized free fatty acids outcompete the hydroperoxides for the HsPrdx3 binding site, thus inhibiting the oxidation of the latter. Can the authors say anything about this issue based on their docking and MD simulations? Eg. does the -OOH group contribute significantly for binding energy?

As indicated above, we do not consider that the binding of the fFA-OOH to HsPrx3 is of high affinity. The molecular docking experiments clearly indicates the preference of these type of substrate to bind to the hydrophobic patch in the proximity of the peroxidatic Cys. Among these conformation clusters, one the best in terms of binding energy and also the most represented one, shows the hydroperoxide group of the fFA-OOH positioned very close to the peroxidatic cysteine, suggesting this conformation as very probable. However, molecular docking binding energy predictions are only indicative in this case, thus we cannot assure that the hydroperoxide group contribute to the binding energy.  We have corrected the discussion section to make clearer these aspects, which were not only addressed by reviewer 2 but also reviewer 3

Line 416: footnote 5 is hidden

It is correct. It was not hidden in the word and pdf file we uploaded in the Journal page, but the word file prepared by the Journal (which was differently formatted) has the footnote 5 hidden, and we have added it again.  It now reads:

“5AA critical micelle concentration ranges between 10-60 μM [70,71].”

Minor

Line 16: Please pay attention to superscripts in the revised version

The reviewer is right, all the superscripts, subscripts and even italics in the Abstract were lost during the submission progress. We corrected it in the revised manuscript.

Line 17, please write the meaning of the PGG2 abbreviation in the abstract

Done. It is now indicated in the abstract “The endoperoxide hydroperoxide PGG2, an intermediate in prostanoid synthesis, oxidized HsPrx3 with similar rate constant, but was less effective in causing hyperoxidization”.

Line 63: Please consider “Both, free and lipid-bound FA-OOHs…”

Done

Line 150: Please consider “DTPA” and define the abbreviation.

Changed.

Line 154: Readers may appreciate some explanation of why EgTR is preferable to other Thioredoxin Reductases for this assay.

The coupled assay used to measure the activity of HsPrx3 during catalysis, consist in the catalytic cycle that we are investigating, the reduction of fatty acid hydroperoxides by thioredoxin 1 catalyzed by HsPrx3 (first cycle), and another cycle, that we only use for easily follow the reaction, which is the reduction of thioredoxin by NADPH catalyzed by TR (second cycle). This second cycle  is not the one under study in the manuscript, but it is necessary for easily measuring the activity of the first cycle, provided the second cycle is not rate limiting the oxidation of NADPH ( the rate of the second cycle is higher  that of the first, which we always prove by determining the effect of TR concentration (which should not affect the rate of NADPH oxidation by fFAOOH in the presence of all the other components (Trx, TR, Prx3) if TR is not rate limiting) and the effect of Prx3 concentration (which should affect the rate of NADPH if the first cycle is rate limiting) at different concentrations of substrates. Bearing that in mind, it is not really important the source of TR, provided it is active using Trx1 as substrate and it is not rate limiting the overall NADPH oxidation by the fatty acid of interest. We used the thioredoxin reductase domain of EgTR which is a selenium containing protein, like mammalian TR, and its purified with high yields and purity in our lab. A brief explanation of this was added in the manuscript, in Methods (2.5 Catalytic activity of HsPrx3).

Line 187. Please define abbreviation “ME” at first mention. Also, a betta appears to be missing, as per the abbreviations list.

The symbols of betta in line 187 were change for another one during submission of the manuscript, and we corrected it in the revised version of the manuscript. However, b-ME was defined in section 2.1 (reagents) and included in the list of abbreviations.

Lines 276-281: Figure 1. Was this experiment repeated to check for reproducibility?
In this and subsequent figures, please pay attention to the layout of the legends, as the text in lines 278-281 (in this case) is formatted as the main text, which may confuse the typesetter.

The experiment was repeated and results were very similar. Thank you very much for indicating that the layout of the legend figure was not clear. In fact, the manuscript that was formatted by the journal changed the font to Palatino linotype font 10 in the main text and Palatino linotype font 9 in the legends of the Figures and also in the Tables. We have checked and in all the figure legends the font is now Palatino linotype font 9. I suppose that it is the style the journal utilizes for the main text and figure legends. We realized that Figure 1 has been cut at the bottom, the name of the x axis which is Time (s) has been cut. It was not cut in the Figures provided as high-quality Tiff images that I am coping below

We are also substituting the image in the new manuscript file.

Table 1: Where there replicates, repeats? How many?

In table 1 we show the initial rates of NADPH consumption of the representative experiment we selected to show in Figure 1. As indicated above, results were qualitative similar in the other experiments.

Line 290: Consider “[…] the first phase was so rapid that it was complete in a few ms […]”

Done

Figure 2: How many experiments and/or replicates were made? What do the dots and error bars in panels C, D represent? Are the dots means/medians from independent experiments or from replicates? Do the error bars represent standard deviations, standard errors of the mean, or something else?

The experiments were repeated 3 times and in the revised version of the manuscript we are informing the mean value of the rate constant and the standard deviation in Table 3. In panels C and D, the dots and the error bars are medians from replicates from the experiment selected to show, we have clarified that in the manuscript text.

Line 361: Consider “Treatment with excess 2-iodoacetamide caused alkylation of reduced HsPrx3 […]”

Done

Lines 369-376: Don’t these observations suggest that oxidation of the sulfinic acid to a sulfonic acid by 15(S)-HpEPE is not much slower than oxidation of the sulfenic acid to the sulfinic acid?

The reviewer might be right in his suggestion. However, since we have not made further attempts to measure the rate constant of oxidation from sulfinic acid to sulfonic acid in addition to this mass spectrometry determinations, we prefer not to include this in the revised manuscript.

Figure 5 and Table 2: Since HsPrdx3 without the targeting sequence has just three Cys residues, the observation of HsPrx3-4CAM needs some explanation.  

The reviewer is right, we have added a sentence to indicate that treatment with IAA can cause, in addition to cysteine alkylation, some side reactions such as the modification of the amino group at N-terminus, as well as those on the side chains of several amino acids (ref Suttapitugsakul S, Xiao H, Smeekens J, Wu R. Mol Biosyst. 2017 21;13(12):2574-2582, now included in the manuscript reference list as ref 73).  In the case of Prxs, the reaction with alkylating agents has been reported to be slow, particularly at the CysP (ref.  Alexander V. Peskin et al, The High Reactivity of Peroxiredoxin 2 with H2O2 Is Not Reflected in Its Reaction with Other Oxidants and Thiol Reagents, Journal of Biological Chemistry, 282, 2007, 11885-11892). We preferred to use a relative high concentration of alkylating agent to promote a fast alkylation of Cys residues, avoiding artifactual thiol group modifications during mass spectrometry experiments, even when we had a minor fraction of side reactions.

Lines 527-530: Considering typical PLA2 and GPX4 activities in mitochondria, just a minor fraction of phospholipid hydroperoxides should be released as fFA-OOHs before the hydroperoxide being reduced via GPX4 [e.g. doi:10.1016/0891-5849(95)00040-5]. Thus, this seems to be an unlikely source of fFA-OOHs capable of significantly hyperoxidizing HsPrdx3 unless GPX4 is downregulated.

In fact, mitochondrial GPx4 expression varies very much among tissues. It has a great expression in testis, but not in most somatic tissues, where cytosolic GPx4 is the isoform more abundant. Please see: Schneider M, Förster H, Boersma A, Seiler A, Wehnes H, Sinowatz F, Neumüller C, Deutsch MJ, Walch A, Hrabé de Angelis M, Wurst W, Ursini F, Roveri A, Maleszewski M, Maiorino M, Conrad M. Mitochondrial glutathione peroxidase 4 disruption causes male infertility FASEB J. 2009 Sep;23(9):3233-42 doi: 10.1096/fj.09-132795. In simulations shown in table 4 and 5, we are considering a somatic tissue with very low expression of GPx4. We are indicating this in the revised version of the manuscript. However, it is true that not all somatic tissues express low concentrations of mitochondrial GPx4, for example it has been recently shown that mGPx4 is highly expressed in photoreceptors.as indicated in Introduction.

Line 551: Consider “reduction of the redoxins […]”

Done

Table 5: The table is somewhat confusing as presented. First, it is mis-formatted, with the columns mis-aligned with the respective headers, which makes it difficult to read. Second, the title is inconsistent with the contents, since what is actually presented are the Prx3-SO2- concentrations, in µM, not the reaction yields, which are dimensionless quantities. And since the main take are the fractions of Prdx3 hyperoxidized in each condition, the message will perhaps be better conveyed by presenting these fractions (by dividing the [Prx3-SO2- ] by the total Prdx3 concentration in each case) instead of [Prx3-SO2-].

We thank the reviewer for his suggestion, and we have modified the Table title and indicate the fractions of HsPrx3 hyperoxidized in the revised version of the manuscript. 

Round 2

Reviewer 2 Report

In the revised version and responses to the reviews on the manuscript “Mitochondrial peroxiredoxin 3 is rapidly oxidized and hyperoxidized by fatty acid hydroperoxides”, the authors convincingly addressed the questions I raised in the previous version. The authors included details and interpretations that indeed clarified some issues of the first version, which certainly strengthened the manuscript. With these additions the novelty and relevance of the findings reported are well highlighted. I therefore consider that it is now suitable for publication in Antioxidants.